# Species traits and network structure predict the success and impacts of pollinator invasions

Fernanda S. Valdovinos [1,2], Eric L. Berlow[3], Pablo Moisset de Espanés[4,5], Rodrigo Ramos-Jiliberto[6,7], Diego P. Vázquez[8,9] & Neo D. Martinez[10]

Species invasions constitute a major and poorly understood threat to plant–pollinator systems. General theory predicting which factors drive species invasion success and subsequent effects on native ecosystems is particularly lacking. We address this problem using a consumer–resource model of adaptive behavior and population dynamics to evaluate the invasion success of alien pollinators into plant–pollinator networks and their impact on native species. We introduce pollinator species with different foraging traits into network models with different levels of species richness, connectance, and nestedness. Among 31 factors tested, including network and alien properties, we find that aliens with high foraging efficiency are the most successful invaders. Networks exhibiting high alien–native diet overlap, fraction of alien-visited plant species, most-generalist plant connectivity, and number of specialist pollinator species are the most impacted by invaders. Our results mimic several disparate observations conducted in the field and potentially elucidate the mechanisms responsible for their variability.

[1] Department of Ecology and Evolutionary Biology, University of Michigan, 1105 North University Ave, Biological Sciences Building, Ann Arbor, MI 48109, USA. [2] Center for the Study of Complex Systems, University of Michigan, Weiser Hall Suite 700, 500 Church St., Ann Arbor, MI 48109, USA. [3] Vibrant Data, San Francisco, CA 94121, USA. [4] Centro de Modelamiento Matemático, Universidad de Chile, Beauchef 850, Santiago 8370397, Chile. [5] Instituto de Dinámica Celular y Biotecnología, Av. Beaucheff 850, Santiago 8370397, Chile. [6] GEMA Center for Genomics, Ecology & Environment, Universidad Mayor, Camino La Pirámide 5750 Huechuraba, Santiago 8580745, Chile. [7] Programas de Postgrado, Facultad de Ciencias, Pontificia Universidad Católica de Valparaíso, Av. Brasil 2950, 2340025 Valparaíso, Chile. [8] Instituto Argentino de Investigaciones de las Zonas Áridas, CONICET, CC 507, 5500 Mendoza, Argentina. [9] Facultad de Ciencias Exactas y Naturales, Universidad Nacional de Cuyo, Padre Jorge Contreras 1300, M5502JMA Mendoza, Argentina. [10] Department of Ecology & Evolutionary Biology, The University of Arizona, Tucson, AZ 85721, USA. Correspondence and requests for materials should be addressed to F.S.V. (email: fsvaldov@umich.edu)

Mutualistic networks of plants and their pollinators are key promoters of terrestrial biodiversity[1] and crucial for our society's food security[2,3]. Unfortunately, the introduction of alien species into native ecosystems, together with climate change, widespread application of pesticides, habitat loss, and degradation severely threatens the integrity of these systems and their critical ecosystem services[2,4]. Despite the importance of plant–pollinator networks, we still lack predictive understanding of the factors and mechanisms determining their invasibility[5,6] and the subsequent effects on native species[7,8]. Fortunately, such understanding of invasions in complex food webs has recently increased[9,10]. Here, we build on those efforts by further developing consumer–resource theory to elucidate the determinants of invasion success and the impacts of alien pollinators in complex plant–pollinator networks.

Several challenges inhibit a better understanding of the processes an properties of the plant–pollinator systems that determine the success and subsequent impacts of alien invaders on native species. For example, the high mobility of pollinators limits the duration of population experiments[11,12] which obscures critically important longer-term impacts of alien species on native communities[13]. Still, researchers have empirically evaluated impacts of one-third of the 80 bee species introduced as pollinators to date[12]. Although most of that evidence is inconclusive[12] and methodologically suspect[11,13], many studies of the honeybee, *Apis mellifera*, illuminate several potential impacts that alien pollinators may exert on native communities. One such impact is increased competition for floral resources inflicted on native pollinators[14–18] though little or no effect on native populations through shared resources is also frequently observed[13,19–21]. Similarly, several studies show that introduced honeybees reduce the reproductive success of native plants[22–25], while others demonstrate that honeybees effectively pollinate native plants[26–29].

Despite the inconclusive and seemingly contradictory results, invasion studies of food webs[10] suggest that the potential effects of alien pollinators on ecological networks may be successfully predicted based on the characteristics of the alien species and its host community. For example, previous theory[13,30] predicts that extraction of substantial amounts of shared limiting resources by aliens may increase the partitioning or decrease the abundance of resources and extirpate native populations. Alternatively, if the resources extracted by the invader are minor in quantity or otherwise not limiting or not shared with natives, native pollinators may be unaffected[13,30]. Regarding plants, theory predicts[30] that alien pollinators may affect native plants negatively or positively depending on whether the aliens act as major pollinators, secondary pollinators, or floral parasites of the plants. Although these predictions constitute important advances, they insufficiently consider the complex networks of plant–pollinator interactions that determine the dynamics of those systems[31–35]. More explicit and quantitative network theory suggests that impacts of pollinators depend on floral resources being shared by pollinator species and also on the effectiveness of plant reproductive services provided by pollinator species[33,34].

To explore these issues here, we use a dynamic consumer–resource approach that incorporates adaptive foraging of pollinators to mechanistically model pollinators' consumption of floral rewards and reproductive services to plant species[33,34]. We simulate species introductions into models of plant–pollinator networks and compare the same networks before and after alien species introductions[9,10]. This comparison provides a more complete understanding of the invasion process and its effects on the native system. Specifically, we address (1) how traits of alien pollinator species determine invasion success, (2) how network structure affects community resistance to species introductions, and (3) how network structure and traits of alien pollinator species interact to determine impacts on native species. We find that species traits of alien pollinators alone predict their introduction success into plant–pollinator networks, while information on the network structure is needed to predict the impact of invading pollinators on native species. Our results mimic several disparate observations conducted in the field and potentially elucidate the mechanisms responsible for their variability.

## Results

**Overview.** We use several common terms to designate the final density of introduced aliens (Methods). "Successful" aliens maintain their density above the pollinators' extinction threshold ($10^{-3}$) through to the end of the simulations. "Unsuccessful" aliens venture below that threshold and are removed from the network. "Naturalized" aliens maintain their density in-between the extinction threshold ($1.0 \times 10^{-3}$) and their initial density ($1.5 \times 10^{-3}$). If aliens increase their density to above 0.5 (i.e., more than 333 times its initial density), they are considered "invaders". No aliens have densities between 0.5 and their initial densities at the end of our simulations. Overall, we find that the abundance of aliens increases with their number of interactions for efficient but not for average foragers (Supplementary Fig. 1).

**Impacts of alien pollinators on native species.** We evaluate the impact of successful aliens on native species in terms of the persistence (i.e., fraction of initial species that persisted through to the end of the simulations), density, and visitation of all species, as well as plant pollination events and floral rewards density at $t = 10,000$ and at $t = 20,000$ (Methods). Our results show that species traits (particularly foraging efficiency) of alien pollinators predict their introduction success into the studied networks, while their impacts on native pollinator and plant species strongly depend on the network structure. Our classification and regression tree (CART, see Methods) analyses indicate that foraging efficiency of alien pollinators is the best predictor explaining 76% and 93% of the variance in pollinators getting naturalized and invading the networks, respectively ($n = 43,200$, Table 1).

**Table 1 Classification and regression tree (CART, $n = 43,200$) analyses of introduction outcomes**

| | Unsuccessful (6%) | Naturalized (46%) | Invader (48%) |
|---|---|---|---|
| Five-folded $R^2$ | 0.85 | 0.91 | 0.98 |
| Main contributors | Pollinator mortality: 19%Alien foraging efficiency: 17%Fr. plant sp visited by alien: 16%Alien mean Jaccardian index: 15%Alien adaptive behavior: 10% | Alien foraging efficiency: 76%Pollinator mortality: 7%Fr. plant sp visited by alien: 6% | Alien foraging efficiency: 93% |

Results as a function of 30 properties of networks and alien pollinator species. Separate analyses were conducted for each of the three binary outcomes (i.e., unsuccessful, naturalized, and invader). The main contributors include only factors that account for >5% of the variance in whether or not the outcome of the introduction fell within the categories identified by the column heading among our 43,200 simulated introductions. The percentages of those simulations falling into the categories are given to the right of the categories in parentheses

**Table 2 Outcomes of introducing alien pollinators into pollination networks**

|  | Average foragers | | Efficient foragers | |
| --- | --- | --- | --- | --- |
|  | All data ($n = 21,600$) | High mortality ($n = 10,800$) | All data ($n = 21,600$) | High mortality ($n = 10,800$) |
| Not established | 11% | 21% | 1% | 2% |
| Naturalized | 89% | 79% | 2% | 4% |
| Invaded | 0% | 0% | 97% | 94% |
| Abundance (s.d.) | $1.3 \times 10^{-3}$ ($10^{-3}$) | $1.2 \times 10^{-3}$ ($10^{-3}$) | 279 (353) | 3.5 (2.1) |

Percentages of simulations falling into categories indicate the fraction of $n$ simulations within the categories. Standard deviations are in parentheses

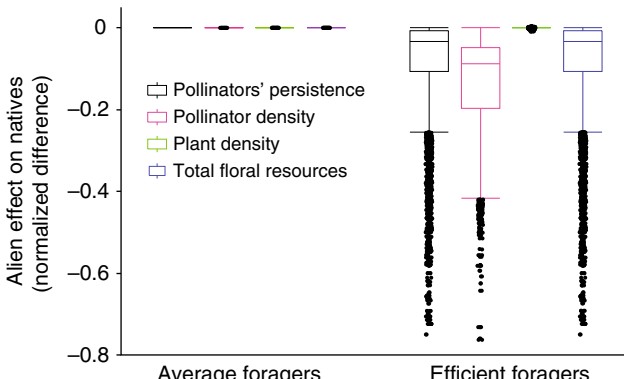

**Fig. 1** Effects of alien pollinators on native species for all network structures. Effects exerted by average foragers (i.e., visitation rates equal to the mean visitation rate of native pollinators; left) were less negative than effects exerted by highly efficient foragers (i.e., visit twice as many flowers per unit time as do native pollinators on average; right). Effects were calculated as the normalized difference of native variables after and before the alien introduction, including (from left to right) persistence and total density of pollinator species, total density of plant species, and total floral resources. Effects displayed separately for alien pollinators that were average (four on the left) and efficient foragers (right) using outlier box plots whose center line, box, and whiskers indicate the median, 75–25% quartiles, and 1.5 * IQR, where IQR = 3rd minus 1st quartile

Successful average foragers (i.e., visitation rates equal to the mean visitation rate of native pollinators, see Methods) only become naturalized, are never invasive (Table 2), and never significantly impact native species (Fig. 1). By contrast, highly efficient foragers (i.e., visit twice as many flowers per unit time as do native pollinators on average, see Methods) virtually always invade the networks (Table 2), but their impacts on native species vary so much across network structures that they cannot be predicted solely based on aliens' foraging efficiency (Fig. 1, see CART results in Supplementary Tables 2 and 3). In fact, generalized linear models show that aliens' foraging efficiency strongly interacts with network properties to determine aliens' impacts on natives (see Table 3 and Supplementary Table 4). Invading pollinators exert the strongest impact on the persistence of native pollinators in networks with high diet overlap between invasive and native pollinators (Fig. 2a: high Jaccardian similarity explains 62% in CART model, five-folded $R^2 = 0.89$, Supplementary Table 2). This finding is corroborated by a generalized linear model showing a highly significant effect of the alien Jaccardian index and its highly significant interaction with the alien foraging efficiency (Table 3). The negative impact of invaders highly overlapping their diets with native pollinators is the strongest in networks where the most generalized plant is pollinated by most of the native pollinators (Fig. 2a: high connectivity of the most-

generalist plant species, color gradient, explains 13% in the same CART model; corroborated by the significant interaction term among alien foraging efficiency, alien Jaccardian index, and connectivity of the most-generalist plant species in Table 3). Invading pollinators predictably decrease the total density of native pollinators in networks where invaders pollinate a high fraction of the plant species, regardless of invaders' diet overlap with the native pollinators (alien's number of interactions explains 99% of the variance in the aliens' effect on the total density of native pollinators in CART model, five-folded $R^2 = 0.99$, Fig. 2b; corroborated by the highly significant interaction term between the alien foraging efficiency and the fraction of plant species visited by the alien pollinator in Table 3). The effects of invading pollinators on plants were much weaker and less consistent than their effects on native pollinators. Invading pollinators left the persistence of native plants unaffected but had a hump-shaped effect on native plant density by positively affecting native plants at low final alien density and negatively affecting native plants at high final alien density (alien density explains 60% of the variance in the aliens' effect on the total density of native plants in CART model, five-folded $R^2 = 0.82$, Fig. 3, Supplementary Table 3). For a given alien density, networks with more native species of specialized pollinators exhibited a stronger decline of plant density after alien invasion (fraction of specialist native pollinators explained 19% of the variation in the aliens' effect on the total density of plants in CART model, Fig. 3; see the highly significant interaction term between the alien foraging efficiency and the fraction of specialist native pollinator species in Table 3).

**Explanatory mechanisms.** Our results indicate that invaders impact native pollinators primarily by strongly decreasing the floral rewards of the plant species the invaders visit. This decrease reduces the persistence of native pollinators in networks with high diet overlap between invasive and native pollinators. This is because increased sharing of links between a native pollinator species and the invader reduces the floral resources available to native species. This negative effect becomes stronger in networks whose most generalized plant species interact with most of the pollinators in the network, including the alien because such high plant generality increases the fraction of native pollinators sharing floral rewards with the invaders. In contrast, aliens that do not visit the most-generalist plant affect native pollinators very little. At the other extreme, native pollinators that share all their links with the invaders always go extinct due to insufficient food. Native pollinators with several links not shared with the invaders usually survive by reassigning their foraging efforts to their plant partners not visited by the invaders. However, the total density of the persistent native pollinators always decreases as the fraction of native plant species visited by the invaders increases regardless of the diet overlap between invaders and natives. This is because native pollinators that survive the invasion reassign their foraging efforts from plants shared with the invaders to plants with fewer

**Table 3 Significant results of generalized linear models (GLM) for the effects of alien pollinators on native species**

|  | Estimate | Std. error | t Value |  |
|---|---|---|---|---|
| *Alien effect on pollinators' persistence (AIC: -5153.8)* |  |  |  |  |
| Alien foraging efficiency (A. For. Eff.) | 0.04 | 0.02 | 2.46 | * |
| Alien mean Jaccardian index (A. Jacc.) | 1.52 | 0.45 | 3.37 | *** |
| Connectivity of the most-generalist plant sp (Conn. Gen. P) | 0.014 | 0.003 | 4.56 | *** |
| **A. For. Eff. * A. Jacc.** | **−1.14** | **0.25** | **−4.63** | *** |
| A. For. Eff. * Conn. Gen. P | −0.007 | 0.002 | −4.31 | *** |
| A. For. Eff. * Fr. plant sp visited by alien | 0.20 | 0.03 | 6.56 | *** |
| A. Jacc.* Conn. Gen. P | −0.14 | 0.05 | −2.64 | ** |
| Conn. Gen. P * Fr. plant sp visited by alien | −0.036 | 0.006 | −6.27 | *** |
| **A. For. Eff. * A. Jacc. * Conn. Gen. P** | **0.10** | **0.03** | **3.43** | *** |
| *Alien effect on native pollinator density (AIC: -11619)* |  |  |  |  |
| Fr. of plant sp visited by alien | −9.5e−3 | 4.7e−3 | −2.03 | * |
| **A. For. Eff. * Fr. of plant sp visited by alien** | **2.6e−2** | **0.2e−2** | **12.9** | *** |
| Conn. Gen. P * Fr. of plant sp visited by alien | −2e−3 | 0.4e−3 | −5.2 | *** |
| *Alien effect on native plant density (AIC: -14201)* |  |  |  |  |
| A. For. Eff. | 3e−3 | 3e−4 | 8.20 | *** |
| Fr. specialist native pollinator sp (Fr. Spec Nat. Pol.) | 2e−3 | 7e−4 | 3.42 | *** |
| **A. For. Eff. * Fr. Spec Nat. Pol.** | **−2e−3** | **5e−4** | **−4.65** | *** |

Predictors of the models were chosen based on CART results and the best models were chosen using AIC. Evaluated predictors for the alien effects on native pollinators' persistence and density were (1) alien foraging efficiency, (2) alien mean Jaccardian index, (3) connectivity of the most-generalist plant species, and (4) fraction of plant sp visited by the alien. Evaluated predictors for the alien effects on plant density were (1) alien foraging efficiency, (2) fraction of plant species visited by the alien, and (3) fraction of specialist pollinator species. Full models are described in Table S4, Supplementary Information. Data used include independent species introduction in 1200 network structures, 600 each for average and efficient alien foragers (see Methods and Supplementary Table 4). The underlined terms are mentioned in the main text. Bold entries are mentioned in the main text. Significance codes for P values: 0 "***", 0.001 "**", and 0.01 "*"

floral rewards than in the plants shared with the invader before the invasion (as empirically shown by[48]). This reduction of rewards available to natives and subsequent reassignment inevitably reduces native pollinators' density.

Invaders either weakly increased or decreased the total density of native plants depending on the balance between the pollination services the plants directly receive from the invaders and the services the plants stop receiving from native pollinators lost due to the invasion. This balance becomes negative when many of the native pollinators contributing to plant reproduction go extinct, which happens at high invaders' density and in networks with many specialist pollinators. At low-to-moderate invader density within networks of few specialist pollinators, invaders weakly increase native plant density by increasing the reproduction of native plants without outcompeting native pollinators that also contribute to the plant reproduction.

## Discussion

Our consumer–resource approach to complex plant–pollinator networks provides a first step to understand and predict invasion success and impacts on native species of alien pollinators based on the aliens' traits, the network structure of plant–pollinator communities, and the adaptive behavior of alien and native pollinators. This approach allowed us to explicitly study mechanisms behind the invasion process of alien pollinators. Studying such mechanisms in the field is very challenging due to the high mobility of pollinator species[11,12].

Our results mimic many of the disparate observations conducted in the field while potentially elucidating the mechanisms that may be responsible for variability (and apparent contradictions) in the empirical results. In particular, our results closely mimic the empirical findings of the honeybee's invasion process and impacts on natives, as well as the displacement of the Patagonian bumblebee by European bumblebees[36] (*Bombus terrestris* and *Bombus ruderatus*). Like the honeybee and European bumblebees, the invaders in our simulations were highly efficient foragers. Our simulations support the empirical findings of (1) European bumblebees outcompeting their native congeners with

very similar niches for floral resources[36], (2) honeybees negatively impacting native pollinators through increased competition for floral resources[14–18], (3) little or no effect of invasive pollinators on native pollinators[13,19–21], and (4) hump-shaped effects of the abundance of alien pollinators on native plants[37]. We find that the network structure of the host community mediates the presence and strength of impacts on native pollinator by an invader like the honeybee. For example, we expect weaker (or no) effects of honeybees on native pollinators in networks where the most-generalist plant is weakly connected and where a low fraction of native plant species is visited by the honeybee. Similarly, our simulations support empirical studies showing that introduced honeybees reduce the reproductive success of native plants[22–25] and also other studies demonstrating that honeybees effectively pollinate native plants[26–28]. Our results suggest a resolution to this apparent contradiction by demonstrating that invasive pollinators can either increase or decrease the reproduction of native plants depending on the invaders' density and the fraction of native pollinators specialize on only one plant. Additionally, our work supports the previously described hump-shaped effects of the abundance of alien pollinators on native plants[37] and suggests that this pattern can be effectively explained with consumer–resource mechanisms operating within a complex network. While floral damage may also reduce floral rewards[37], our results suggest that such damage is not necessary. Finally, our results do not support previous theoretical work, suggesting that pollinator species indirectly benefit each other by sharing mutualistic partners[32] (i.e., plants). Our results rather support the absence of such positive effects in empirical records[12]. Such indirect positive effects among pollinator species do not occur in our model because the depletion of floral rewards strongly decreases native pollinator density even when aliens increase the density of plants pollinated by those natives. Still, pollinator density only increases plant density until processes in the plant life cycle other than pollination such as seedling recruitment or adult survival limit plant density.

Our study may help to productively focus empirical research on a few of many different factors influencing invasion success and impacts of alien pollinators. These few factors are aliens' foraging

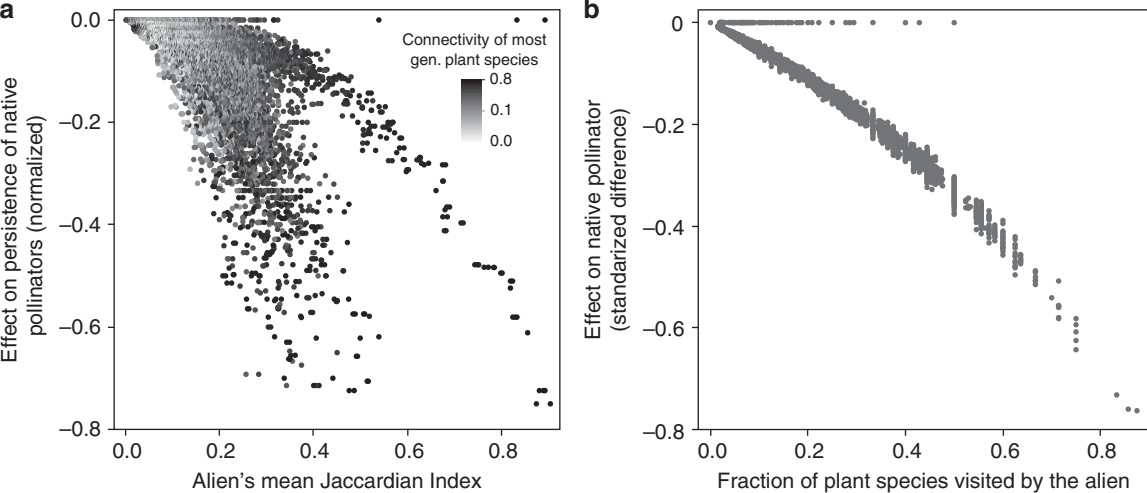

**Fig. 2** Top factors determining the effect of alien efficient foragers ($n = 21,600$) on the persistence and density of native pollinators. In **a**, the alien's mean Jaccardian index (i.e., diet overlap with native pollinators; x axis) and the connectivity of the most-generalist plant species (i.e., fraction of total native links connected to the most-generalist plant species; grayscale normalized as percentile) contributed 62% and 13%, respectively, to the CART model (five-folded $R^2 = 0.89$, see Supplementary Table 2) predicting the alien effect on native pollinator persistence, and were also highly significant factors in generalized linear models (Table 3). In **b**, the fraction of native plant species visited by the alien contributed 99% to the CART model (five-folded $R^2 = 0.99$, see Supplementary Table 2) predicting the alien effect on native pollinator density, and was also a significant factor in the generalized linear model (Table 3). Effects were calculated as the normalized difference of native persistence and density after and before the alien introduction. Zero and close-to-zero effects correspond to unsuccessful (1%) and naturalized (2%) efficient foragers. All effects are negative or zero

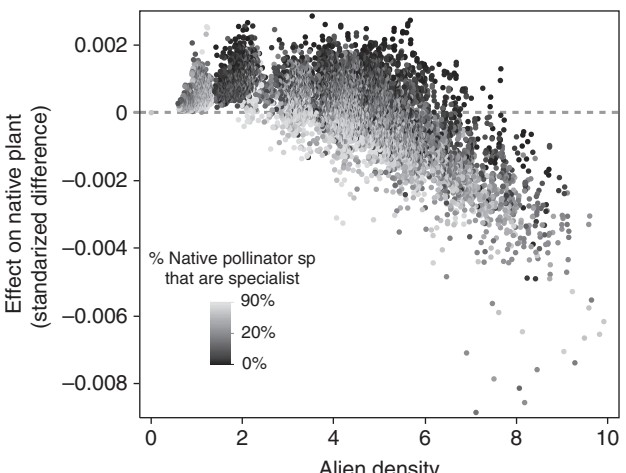

**Fig. 3** Top factors determining the effect of alien efficient foragers at high pollinators' mortality rates ($n = 10,800$) on the density of native plants. The alien density (x axis) and the fraction of native pollinator species that are specialist (i.e., one link; grayscale normalized as percentile) explained 60% and 19% of the variance in plant density, respectively in CART models (five-folded $R^2 = 0.82$, see Supplementary Table 3). Invasive pollinators did not cause any plant extinction and their effect on plant density was negligible for low pollinators' mortality rates

efficiency, diet overlap with native pollinators, fraction of plant species visited by the alien pollinator, connectivity of the most-generalist plant in the host community, and fraction of native pollinators with only one interaction. Such focus informs trait-based prediction in ecology[38] by suggesting that explanations based on species' traits alone may be limited. In particular, our results show that traits of alien species (foraging efficiency) predict the invasion success of alien pollinators, while traits of networks are needed to predict the impact of invading pollinators on native species. Research that continues this trend of incorporating bio-logical processes such as consumer–resource interactions,

reproductive services, and species traits into the study of ecological networks may further help scientists discover other informative, surprising, and profoundly counterintuitive behaviors of complex ecological networks.

## Methods

**Network structures**. Following previous dynamical studies of ecological network dynamics[34,39], we distinguish two fundamental components: the structure of the networks and the dynamics occurring in those networks. The structure of a network broadly describes which links are present or absent between all plant and pollinator species in a system irrespective of the strength of the link. The dynamics occurring within plant–pollinator networks consist of changes in the abundance of the interacting species and/or the strength of the interactions, that is, changes in the values of the nodes and/or links, respectively. The network structures we used are those of 1200 networks previously generated[34] using Thèbault and Fontaine's stochastic algorithm[39]. This algorithm randomly and independently assigns each plant $i$ and pollinator (animal) $j$ the respective interaction probabilities of $P_{Pi}$ and $P_{Ai}$ drawn from a power-law distribution of degree −2. This creates relatively few generalist species that interact with many other species and many specialist species that interact with few other species as is typically seen in empirical networks[31]. Then, with a probability "pnest", each species' interacting partners are sequentially chosen from all potential partners with a probability $\frac{P_{Pi}}{\sum_{k=1}^{S^P} P_{Pk}} \left( or \ \frac{P_{Aj}}{\sum_{l=1}^{S^A} P_{Al}} \right)$. With a

probability of 1-pnest, partners are chosen with a probability $1/S^P$ (or $1/S^A$), where $S^P$ (or $S^A$) indicate the number of plant (or animal) species. When "pnest" is high, the algorithm generates more-nested networks.

Our generated networks exhibit empirically observed patterns of species richness ($S$) inversely varying with connectance ($C = L/(S^A * S^P)$ where $L$ = number of links). This pattern was followed by stochastically generating three sets of 400 networks with each set broadly centered at three combinations of $S$ and $C$: $S = 40$ and $C = 0.25$, $S = 90$ and $C = 0.15$, and $S = 200$ and $C = 0.06$. Half of the networks within each set were significantly nested. Nestedness of these networks (NODFst[40]) varies from −0.33 to 2.3 which includes the empirically observed range of nestedness (−0.37 to 1.3, ref. [34]). The ratio of the number of animal to plant species ($S^A/S^P$) also matched the empirically observed mean[41] of ~2.5.

**Network dynamics**. We simulated the dynamics within our 1200 networks using Valdovinos et al.'s consumer–resource model[34] of population dynamics with adaptive foraging. This model describes the population dynamics of each plant and animal species, the dynamics of the total floral rewards of each plant species, and the adaptive dynamics of the per-capita foraging preferences of each pollinator species for each plant species. The model parameters are described below and in Supplementary Table 1 along with their units. The model calculates the change of the density ($p_i$) of plant individuals, each with a single flower, of species $i$ over time

as

$$\frac{dp_i}{dt} = \gamma_i \sum_{j \in A} e_{ij} \sigma_{ij} V_{ij} - \mu_i^P p_i \qquad (1)$$

where the first and second terms on the right represent population gains and losses, respectively. The realized fraction of seeds that recruit to adults is $\gamma_i$

$$\gamma_i = g_i \left( 1 - \sum_{l \neq i \in P} u_l p_l - w_i p_i \right) \qquad (2)$$

where $g_i$ is the maximum fraction of seeds that can recruit to adulthood. We subject $g_i$ to both interspecific ($u_l$) and intraspecific ($w_i$) competition with $u_l < w_i$. $e_{ij}$ in Eq. (1) is the constant expected number of seeds produced by a pollination event. We address the impacts of pollinator sharing on plant fitness by calculating $\sigma_{ij}$, the fraction of visits of animal $j$ to plant $i$ that successfully pollinate plant $i$

$$\sigma_{ij} = \frac{\varepsilon_i V_{ij}}{\sum_{k \in P_j} \varepsilon_k V_{kj}} \qquad (3)$$

where $\varepsilon_i$ is the pollen production of plant $i$ and $V_{ij}$ is the frequency of visits by animal species $j$ to plant species $i$

$$V_{ij} = \alpha_{ij} \tau_{ij} a_j p_i \qquad (4)$$

where $V_{ij} = 0$ if plant $i$ and animal $j$ do not interact. The dimensionless function discussed further below, $0 \leq \alpha_{ij} \leq 1$, is the foraging preference of pollinator $j$ on plant $i$. $\tau_{ij}$ is the pollinator's visitation efficiency on plant $i$, which corrects for units and is fixed at 1 in this study. $\mu_i^P$ in Eq. (1) is the constant density-independent per-capita mortality rate of plant $i$.

The change of the density of pollinator individuals ($a_j$) of species $j$ over time is

$$\frac{da_j}{dt} = \sum_{i \in P} c_{ij} V_{ij} b_{ij} \frac{R_i}{p_i} - \mu_j^A a_j \qquad (5)$$

where $c_{ij}$ represents the constant per-capita conversion efficiency of pollinator $j$ converting plant $i$'s floral resources into $j$'s births. $b_{ij}$ is the constant efficiency of pollinator $j$ extracting plant $i$'s floral resources ($R_i$) whose change over time is

$$\frac{dR_i}{dt} = \beta_i p_i - \varphi_i R_i - \sum_{j \in A_i} V_{ij} b_{ij} \frac{R_i}{p_i} \qquad (6)$$

where $\beta_i$ is plant $i$'s per-capita resource production rate and $\phi_i$ is a constant self-limitation parameter. $\mu_j^A$ in Eq. (5) is pollinator $j$'s constant density-independent per-capita mortality rates.

Adaptation of pollinator $j$'s foraging preference on plant $i$ ($\alpha_{ij}$ in Eq. (4)) is

$$\frac{d\alpha_{ij}}{dt} = G_j \alpha_{ij} \left( c_{ij} \tau_{ij} b_{ij} R_i - \sum_{k \in P_j} \alpha_{kj} c_{kj} \tau_{kj} b_{kj} R_k \right) \qquad (7)$$

where $G_j$ is the basal adaptation rate of foraging preference and $\Sigma \alpha_{ij} = 1$ for all plants that each pollinator $j$ visits. Pollinator $j$ allocates more foraging effort to plant $i$ whenever such reallocation enhances $j$'s food intake.

The parameters for plants, including competition coefficients, floral reward productivity, and mortality stochastically vary among plant species within 10% of constant values. The non-structural parameters for constraining pollinator dynamics, including visitation efficiency and mortality do not vary among native pollinator species. This parameter choice allowed us to disentangle the effect of network structure from the population dynamics of native pollinators, which were more sensitive to invasions than plants. We ran our model for each of our 1200 networks for 10,000 time steps and then measured several topological properties and dynamic variables described below as response variables. At $t = 10,000$, we stopped the simulations to introduce an alien species and then ran the model for another 10,000 time steps, after which we remeasured the response variables. Most networks achieve stable equilibrium at around 3000 time steps. Running the model for longer ensures that transient dynamics minimally affect the differences between the network dynamics before and after introductions. Sensitivity analyses of the dynamic model have been performed in previous studies[34,39], and the main results presented here (i.e., foraging efficiency of alien pollinators is sufficient to predict invasion success, while information on the network structure is also required to predict the invaders' impact on natives) are qualitatively robust to variation in parameter values.

**Pollinator species introductions**. The "alien" pollinator species were added into the network at $t = 10,000$ with an initial density of $1.5 \times 10^{-3}$. This is slightly above the extinction threshold for all pollinators in the network ($1 \times 10^{-3}$). Except for adaptive foraging, alien species' parameters were assigned as the average of natives,

unless otherwise stated. All native pollinators visiting more than one plant species adaptively forage, while only aliens designated as such adaptively forage. The number of links between alien pollinators and native plants were assigned as the mean of the 30% most specialized or generalized pollinators depending on whether the alien's foraging breadth was specified as specialist or generalist. This arbitrary choice produced alien pollinators visiting between 1 and 13 plant species. The plant species visited by alien pollinators were chosen using one of three different algorithms that linked the alien to (1) the most-connected species, (2) the least-connected species, and (3) randomly selected species independent of the natives' number of links. Six types of alien pollinator species were generated corresponding to two levels of three different properties: generality (the number of plant species they pollinate), visitation efficiency (the number of flowers visited per unit time), and adaptive foraging (ability to prefer plant species within the pollinator's diet with higher-than-average rewards). Specialists pollinate only one plant species and therefore cannot adaptively forage and are not included in adaptive foraging types of invaders. These six types are 1 specialists, 2 efficiently visiting specialists, 3 generalists, 4 efficiently visiting generalists, 5 adaptively foraging generalists, and 6 adaptively foraging and efficiently visiting generalists. Efficiently visiting alien species of types 2, 4, and 6 visit twice as many flowers per unit time as do native species on average. The visitation rates of type 1, 3, and 5 aliens equal the mean rate of native species. The foraging preference for each plant species visited of adaptively foraging types 5 and 6 of aliens was initially assigned to be one divided by the number of species the aliens visited and then allowed to increase (decrease) in response to a plant species having more (less) abundant floral rewards than the mean reward availability among its partners. Foraging preferences of non-adaptive type 1–4 aliens were initially set as were adaptive foraging aliens but remained fixed at those preferences. All simulations were run at two different mortality rates (0.05 and 0.001) for all pollinators, including the introduced alien to explore systems in which it was more or less challenging for pollinators to persist within their communities. Overall, our combination of introducing six types of aliens into 1200 native networks subjected to two different mortality scenarios according to three different attachment algorithms resulted in a total of 43,200 ($6 \times 1200 \times 2 \times 3$) simulated introductions. Our simulation design includes stochasticity by drawing parameter values from uniform distributions for the dynamic model (see Table S1) and by randomly drawing the plant species that the alien pollinator will visit following the rules described above.

**Response variables and statistical analysis**. We use several common terms to designate the final density of introduced aliens. "Successful" aliens maintained their density remaining above the pollinators' extinction threshold ($10^{-3}$) through to the end of the simulations. "Unsuccessful" aliens ventured below that threshold. "Naturalized" aliens maintained their density in-between the extinction threshold ($1.0 \times 10^{-3}$) and their initial density ($1.5 \times 10^{-3}$). If aliens increased their density to above 0.5, they were considered "invaders". No aliens had densities between 0.5 and their initial densities at the end of our simulations. We evaluated the impact of successful aliens on native species in terms of the persistence (i.e., fraction of initial species that persisted through to the end of the simulations), density, and visitation of all species, as well as plant pollination events and floral rewards density at $t = 10,000$ and at $t = 20,000$. We measured the normalized effects ($E$) of the aliens on natives as $E = [(A-B)/(A + B)]$ where $A$ is the properties of native species after the introduction and $B$ is the properties of native species before the introduction. The sum in $E$'s quotient reduces bias caused when density increases from extremely low values before introductions result in exceedingly large effects.

To evaluate how the properties of networks and aliens may determine aliens' success and effects on native species, we measured 23 topological properties of each network at $t = 10,000$ and eight properties of each introduced species. Network properties include species richness ($S$), the ratio of plant to animal species ($S^A/S^P$), four measures of linkage density [connectance ($C$), links per species ($L/S$), links per plant species ($L/S^P$), and links per animal species ($L/S^A$)], ten measures of degree heterogeneity [five for plants and five for animals, including the power-law exponent of the degree distributions[41], the percentage of species with one link, the fraction of total links connected to the most-generalist species, mean degree of generalists (30% of species with the most links), and the standard deviations of animal generality and plant vulnerability sensu[42]], four measures of qualitative interaction overlap [maxima and means of Jaccardian[43] similarity indices for animal and plant species], two measures of quantitative interaction overlap [maximum and mean of Horn's index[44] for the foraging preferences of animal species], and nestedness (NODFst[40]). Properties of aliens that we recorded include foraging efficiency, number of interactions, maxima and means of Jaccardian and Horn's similarity indices, density at $t = 20,000$, and whether the alien was an adaptive forager. We used CART[45] to predict introduction success and the aliens' effects on native species using these 31 properties. To evaluate interactions among predictors, we also evaluated "generalized linear models" predicting alien's effects on natives. We built such models using CART results for choosing the model predictors (among the 31 factors here evaluated) and chose the best models using the Akaike information criterion (AIC, see Table S4 in Supplementary Information). We fit the generalized linear models to a subset of our data representing 1200 independent introduction trials (600 each for average and efficient alien foragers, into 100 nested and 100 non-nested networks of each three

richness/connectance combinations described above), for the high-mortality scenario, assuming all aliens are adaptive foragers and get randomly connected to plant species independent of the plants' number of links. CART analyses were conducted in JMP[46] using fivefold cross-validation to avoid overfitting, while generalized linear models were fitted using the "glm" function in R[47] with the default settings for the Gaussian family.

**Data availability**. The computer code used to run all the simulations in this work can be accessed at the repository https://github.com/fsvaldovinos/Pollinator_Invasions.

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

## Acknowledgements

This work was supported by the University of Michigan (to F.S.V.); the US NSF (ICER-131383 and DEB-1241253 (to N.D.M.), US DOE DE-SC0016247 (to NDM), and FONDECYT 1120958 (to P.M.d.E. and R.R.-J.).

## Author contributions

F.S.V. conceived and designed the study and performed the simulations; F.S.V. and P.M.d.E. developed the code; F.S.V. and E.L.B. conducted the statistical analyses; F.S.V., E.L.B., and N.D.M. analyzed the results and wrote the manuscript; and R.R.-J. and D.P.V. provided conceptual guidance.

**Additional information**

**Competing interests:** The authors declare no competing interests.

