## [Peer Review File · Nature Communications]

Reviewers' comments:

Reviewer #1 (Remarks to the Author):

This paper addresses the very interesting and timely question of which factors are driving the invasion success of alien pollinator species in complex resource - pollinator networks. The authors simulate the pollinator - resource dynamics over time using a dynamical model developed a few years ago by the authors themselves. The main results are that foraging efficiency of the alien pollinator predict their invasion success but that the impact the invasion has on the native community is dependent on the structural properties of the network. The paper is clearly written, addresses an interesting topic and also has an important message. However, I have two comments/suggestions that I really think would improve and strengthen the paper.

The results are really interesting pointing at the combined importance of traits of the alien pollinator species and properties of the invading network (including trait space of the native pollinator species); for example alien pollinators with a certain trait (high foraging efficiency) are always successful in invading the network, but their impact on the native community is dependent of the properties of the network. In order to disentangle the effects of the different parameters the authors use a Classification Tree analyses. My question is though if it would not be more interesting to go into more depth with the statistical analyses, in particular looking for interactions among different terms. The interactions between network properties and traits of the alien pollinator would be of particular interest. One suggestion is to use a model selection approach and, due to the large number of parameters to fit, build up from very simple to more complex models that then would include interactions. I think this would indeed strengthen the message of this paper. (One additional effect of identifying potential interacting effects would also be that it would allow the authors to elaborate more in their text on the importance of network structure which could be valuable.)

My second comment is regarding the consumer-resource-model used. This model requires that numerous parameters are described both for the resources and the pollinators. I know the authors in earlier papers performed sensitivity analyses and was slightly surprised this was not included here. It might be that those analyses could be used here as well, but if so at the very least I think the authors should address this in the method section in the main text.

Other than that, I really enjoyed reading the paper and think it would be of interest for a wide audience.

Reviewer #2 (Remarks to the Author):

The authors present their work on computer generated simulations of invasive pollinators into pollination networks. One criticism that could be raised against this work is that it is entirely synthetic in nature, with no "real life" experimental data. However, I would argue that this is precisely the way to investigate the impact on large and complex networks, where real experiments are pretty much impossible. I therefore commend the authors for their work and believe this is a great step forward in this area and am pleased to see them highlight the types of things other researchers should be looking for in their field work.

Minor comments:

Page 6, line 116 (and page 14, line 321). I think the authors meant to put 1.5 instead of 0.5
Page 13, line 285. The authors talk about a native pollinator having more than one partner. I'm assuming they're talking about the number plants they pollinate? Please clarify in the text.

Page 13, lines 286-289. The authors have chosen, what appears to be quite extreme ends of the spectrum from generalist to specialist (30% at each end of the spectrum). What was the actual mean values used? Why choose 30% (it's OK if that was an arbitrary choice, just explain if that was the case or not)?

Page 14, lines 311-314. So it sounds like there was no stochasticity in any one type of invader being introduced to one type of network. Would that simulation always end in the same result? It's OK if that is the case, but if there was stochasticity, why not introduce the same species repeatedly into the same network to look at the range of outcomes?

Reviewer #3 (Remarks to the Author):

The authors establish a mathematical model to study the impact of invading pollinator species in plant pollinator networks. Using extensive numerical simulations over large ensembles of networks of different size and connectance, they study the influence of pollinator foraging traits and network structure for invasion success and impact on native communities.

The study is technically performed at a high level, it includes impressive numerical simulations and a comprehensive analysis of all possible cofactors influencing the numerical outcomes. The methods and results are clearly presented. Thus, it is a study that deserves being published in a major ecological journal.

I have second thoughts, however, if this study is suitable for Nature Communications.

Even though the model setup and the results are undoubtedly novel, most findings are not really astonishing. To give some examples, major findings such as "invaders impact native pollinators primarily by strongly decreasing the floral rewards of the plant species that invaders visit", "sharing of links between a native pollinator species and the invader reduces the floral resources", "these effect becomes stronger in networks whose most generalized plant species interacts with most of the pollinators in the network", etc. would be the first ideas that come to mind even without numerical simulations. That is not to say that I find the study superfluous. In contrast, the differentiated and quantitative analysis of the many factors influencing invasion success and impact definitely has many merits and provides many opportunities for future studies. But, I am afraid that this paper does not provide a real breakthrough and will have only a minor influence on thinking in the field. What could have brought this paper over the top would be a strong connection of simulation results to empirical data (even though I can understand that this may not (yet) be possible for several reasons).

In summary, I recommend to resubmit this paper to a strong ecological journal.

Response to the referees' comments on manuscript NCOMMS-18-03578-T

Please see below our point-by-point response to each reviewer's comment. Our responses are in italic-bold font here and the edits to the manuscript are yellow-highlighted in the main text.

Reviewer #1 (Remarks to the Author):

This paper addresses the very interesting and timely question of which factors are driving the invasion success of alien pollinator species in complex resource - pollinator networks. The authors simulate the pollinator - resource dynamics over time using a dynamical model developed a few years ago by the authors themselves. The main results are that foraging efficiency of the alien pollinator predict their invasion success but that the impact the invasion has on the native community is dependent on the structural properties of the network. The paper is clearly written, addresses an interesting topic and also has an important message.

We thank the reviewer for the interest and time spent on our manuscript, and for the positive comments on our main results, narrative and relevance of our study.

However, I have two comments/suggestions that I really think would improve and strengthen the paper.

We thank the reviewer for these excellent suggestions that greatly improved our manuscript. We believe have thoroughly addressed them.

The results are really interesting pointing at the combined importance of traits of the alien pollinator species and properties of the invading network (including trait space of the native pollinator species); for example alien pollinators with a certain trait (high foraging efficiency) are always successful in invading the network, but their impact on the native community is dependent of the properties of the network. In order to disentangle the effects of the different parameters the authors use a Classification Tree analyses. My question is though if it

would not be more interesting to go into more depth with the statistical analyses, in particular looking for interactions among different terms. The interactions between network properties and traits of the alien pollinator would be of particular interest. One suggestion is to use a model selection approach and, due to the large number of parameters to fit, build up from very simple to more complex models that then would include interactions. I think this would indeed strengthen the message of this paper. (One additional effect of identifying potential interacting effects would also be that it would allow the authors to elaborate more in their text on the importance of network structure which could be valuable.)

Following the reviewer suggestion, we have extended our statistical analysis to evaluate interactions among predictors with particular emphasis on interactions between alien traits and network properties. For that, we evaluated generalized linear models (GLM) predicting alien's effects on natives. We built such models using our CART results for choosing the model predictors (among the 31 factors here evaluated) and chose the best models using the Akaike information criterion (AIC). See added text at the end of the Methods section (L369-380).

Very satisfactorily, these models corroborated and better demonstrate what we tried to show visually in our Figs. 1-3. See added Table 3 in the main text for the significant terms of our GLMs and the added Table S4 in the Supplementary Information for the full models.

In particular, our GLMs show that alien foraging efficiency interacts with the alien jaccard index and the connectivity of the most generalized plant species in the network to determine the alien effect on species persistence of native pollinators, which also corroborates what we show in Fig. 2a (see added text in the Results Section, L139-141 and L145-147). Also corroborated and extended our results shown in Fig 2b (see added text in the Results Section L150-152) and our results shown in Fig 3 (see added text L160-162).

Finally, to better show the variability of aliens' impacts on natives that is analyzed using our GLMs, we replaced our previous Fig 1 depicting means and standard errors by our new Fig. 1 depicting Outlier Box Plots.

My second comment is regarding the consumer-resource-model used. This model requires that numerous parameters are described both for the resources and the pollinators. I know the authors in earlier papers performed sensitivity analyses and was slightly surprised this was not included here. It might be that those analyses could be used here as well, but if so at the very least I think the authors should address this in the method section in the main text.

We thank the reviewer for pointing out this important subject. The revised version of our methods (L290-294) now includes: "Sensitivity analyses of the dynamic model have been performed in previous studies^{34,39}, and the main results presented here (i.e. foraging efficiency of alien pollinators is sufficient to predict invasion success, while information on the network structure is also required to predict the invaders' impact on natives) are qualitatively robust to variation in parameter values."

Other than that, I really enjoyed reading the paper and think it would be of interest for a wide audience.

Reviewer #2 (Remarks to the Author):

The authors present their work on computer generated simulations of invasive pollinators into pollination networks. One criticism that could be raised against this work is that it is entirely synthetic in nature, with no "real life" experimental data. However, I would argue that this is precisely the way to investigate the impact on large and complex networks, where real experiments are pretty much impossible. I therefore commend the

authors for their work and believe this is a great step forward in this area and am pleased to see them highlight the types of things other researchers should be looking for in their field work.

We thank the reviewer for the positive comments on our work, and for the time spent reviewing our manuscript

Minor comments:

Page 6, line 116 (and page 14, line 321). I think the authors meant to put 1.5 instead of 0.5

We actually meant 0.5. We see how all these density values can be confusing, so we clarify: “If aliens increased their density to above 0.5 (i.e. more than 333 times its initial density), they were considered ‘invaders’.” (new L116-117)

Page 13, line 285. The authors talk about a native pollinator having more than one partner. I'm assuming they're talking about the number plants they pollinate? Please clarify in the text.

We thank the reviewer for catching this lack of clarity in our description. The revised version now states “All native pollinators visiting more than one plant species adaptively forage while only aliens designated as such adaptively forage” (new L301)

Page 13, lines 286-289. The authors have chosen, what appears to be quite extreme ends of the spectrum from generalist to specialist (30% at each end of the spectrum). What was the actual mean values used? Why choose 30% (it's OK if that was an arbitrary choice, just explain if that was the case or not)?

We thank for suggesting this valuable piece of information. We added it by stating: “This arbitrary choice produced alien pollinators visiting between 1 and 13 plant species.” (new L305-306)

Page 14, lines 311-314. So it sounds like there was no stochasticity in any one type of invader being introduced to one type of network. Would that simulation always end in the same result? It's OK if that is the case, but if there was stochasticity, why not introduce the same species repeatedly into the same network to look at the range of outcomes?

Our simulation design includes stochasticity in drawing parameter values for the dynamic model and in drawing the plant species that the alien pollinator visit. Therefore, the reviewer is right that an alternative design could have been introducing the same species repeatedly into the same network. However, that alternative design would have created pseudo-replicates for what we wanted to evaluate: the relative contribution of alien traits and network structure in determining the invasion success and subsequent impacts on native species.

We clarify this by stating: “Our simulation design includes stochasticity by drawing parameter values from uniform distributions for the dynamic model (see Table S1) and by randomly drawing the plant species that the alien pollinator will visit following the rules described above” (new L331-334).

Reviewer #3 (Remarks to the Author):

The authors establish a mathematical model to study the impact of invading pollinator species in plant pollinator networks. Using extensive numerical simulations over large ensembles of networks of different size and connectance, they study the influence of pollinator foraging traits and network structure for invasion success and impact on native communities.

The study is technically performed at a high level, it includes impressive numerical simulations and a comprehensive analysis of all possible cofactors influencing the numerical outcomes. The methods and results are clearly presented. Thus, it is a study that deserves being published in a major ecological journal.

We thank the reviewer for the time spent on reviewing our manuscript and positive comments on the quality of our work.

I have second thoughts, however, if this study is suitable for Nature Communications. Even though the model setup and the results are undoubtedly novel, most findings are not really astonishing.

To give some examples, major findings such as "invaders impact native pollinators primarily by strongly decreasing the floral rewards of the plant species that invaders visit", "sharing of links between a native pollinator species and the invader reduces the floral resources", "these effect becomes stronger in networks whose most generalized plant species interacts with most of the pollinators in the network", etc. would be the first ideas that come to mind even without numerical simulations. That is not to say that I find the study superfluous. In contrast, the differentiated and quantitative analysis of the many factors influencing invasion success and impact definitely has many merits and provides many opportunities for future studies. But, I am afraid that this paper does not provide a real breakthrough and will have only a minor influence on thinking in the field.

What could have brought this paper over the top would be a strong connection of simulation results to empirical data (even though I can understand that this may not (yet) be possible for several reasons).

Unfortunately, such data is not available given immense practical challenges (as we described at the beginning of our manuscript). We see our work as an important step towards understanding and predicting invasions success and consequent impacts on native species in plant-pollinator networks, despite those practical challenges. Moreover, our results mimic several disparate observations conducted in the field and potentially elucidate the mechanisms responsible for their variability.

We believe the revised version of our manuscript incorporating the excellent suggestions of the other two reviewers make our results even more compelling for a general audience.

In summary, I recommend to resubmit this paper to a strong ecological journal.

REVIEWERS' COMMENTS:

Reviewer #1 (Remarks to the Author):

The authors did address all comments to satisfactory, and I have nothing further to add.

Reviewer #2 (Remarks to the Author):

The authors have now adequately addressed all comments from the reviewers